# Molecular Mechanisms and Physiological Changes behind Benign Tracheal and Subglottic Stenosis in Adults

**DOI:** 10.3390/ijms23052421

**Published:** 2022-02-22

**Authors:** Alessandro Marchioni, Roberto Tonelli, Alessandro Andreani, Gaia Francesca Cappiello, Matteo Fermi, Fabiana Trentacosti, Ivana Castaniere, Riccardo Fantini, Luca Tabbì, Dario Andrisani, Filippo Gozzi, Giulia Bruzzi, Linda Manicardi, Antonio Moretti, Serena Baroncini, Anna Valeria Samarelli, Massimo Pinelli, Giorgio De Santis, Alessandro Stefani, Daniele Marchioni, Francesco Mattioli, Enrico Clini

**Affiliations:** 1Respiratory Diseases Unit, Department of Medical and Surgical Sciences, University of Modena Reggio Emilia, University Hospital of Modena, 41121 Modena, Italy; alessandreani@yahoo.it (A.A.); gaia.cappiello@gmail.com (G.F.C.); fabianatrentacosti@gmail.com (F.T.); ivana_castaniere@icloud.com (I.C.); fantini.riccardo@yahoo.it (R.F.); lucatabbi@gmail.com (L.T.); darioandrisani@libero.it (D.A.); fillo.gzz@gmail.com (F.G.); giulibru92@gmail.com (G.B.); linda.manicardi3@gmail.com (L.M.); antomor93@hotmail.it (A.M.); serena.baroncini@gmail.com (S.B.); annavaleria.samarelli@unimore.it (A.V.S.); enrico.clini@unimore.it (E.C.); 2Clinical and Experimental Medicine PhD Program, University of Modena Reggio Emilia, 41121 Modena, Italy; 3Otolaryngology Unit, University Hospital of Modena, 41121 Modena, Italy; matteo.fermi3@unibo.it (M.F.); daniele.marchioni@unimore.it (D.M.); franz318@hotmail.com (F.M.); 4Plastic Surgery Unit, University Hospital of Modena, 41121 Modena, Italy; pinelli1massimo@gmail.com (M.P.); giorgio.desantis@unimore.it (G.D.S.); 5Thoracic Surgery Unit, University Hospital of Modena, 41121 Modena, Italy; alessandro.stefani@unimore.it

**Keywords:** subglottic stenosis, tracheal stenosis, relapsing polychondritis, granulomatosis with polyangiitis, web-like stenosis, tracheostomy

## Abstract

Laryngotracheal stenosis (LTS) is a complex and heterogeneous disease whose pathogenesis remains unclear. LTS is considered to be the result of aberrant wound-healing process that leads to fibrotic scarring, originating from different aetiology. Although iatrogenic aetiology is the main cause of subglottic or tracheal stenosis, also autoimmune and infectious diseases may be involved in causing LTS. Furthermore, fibrotic obstruction in the anatomic region under the glottis can also be diagnosed without apparent aetiology after a comprehensive workup; in this case, the pathological process is called idiopathic subglottic stenosis (iSGS). So far, the laryngotracheal scar resulting from airway injury due to different diseases was considered as inert tissue requiring surgical removal to restore airway patency. However, this assumption has recently been revised by regarding the tracheal scarring process as a fibroinflammatory event due to immunological alteration, similar to other fibrotic diseases. Recent acquisitions suggest that different factors, such as growth factors, cytokines, altered fibroblast function and genetic susceptibility, can all interact in a complex way leading to aberrant and fibrotic wound healing after an insult that acts as a trigger. However, also physiological derangement due to LTS could play a role in promoting dysregulated response to laryngo-tracheal mucosal injury, through biomechanical stress and mechanotransduction activation. The aim of this narrative review is to present the state-of-the-art knowledge regarding molecular mechanisms, as well as mechanical and physio-pathological features behind LTS.

## 1. Introduction

Benign laryngotracheal stenosis (LTS) is an umbrella term for various pathological conditions resulting in a narrowing of the airways at the level of the glottis, the subglottic space, or the trachea. Although the aetiology of LTS is mainly iatrogenic (i.e., post-intubation stenosis or cartilage injury after tracheotomy), autoimmune diseases can also be associated with subglottic mucosal inflammatory changes resulting in central airway stenosis (including granulomatosis with polyangiitis, relapsing polychondritis, sarcoidosis and IgG4-related disease) [1]. 

Clinically, severe LTS results in respiratory functional impairment, often challenging to be managed appropriately.

Laryngotracheal surgery may be considered an option for LTS, despite the intrinsic limitations related to the surgical procedure itself and/or the patient’s clinical status [2]. Recently, less invasive techniques such as endoscopy have become more and more popular, even if burdened by frequent relapses requiring reoperation [3]. 

The understanding of the pathophysiology of LTS is still an ongoing process and is crucial to develop new more effective therapeutic strategies. Further, the prevention of LTS and its recurrence represents an important clinical need. Until recently the laryngotracheal scar resulting from airway injury was considered an inert tissue requiring surgical removal to restore airway patency. This assumption has now been revised: tracheal scarring is seen as a fibroinflammatory event triggered by immunological alterations [1]. Studies in animal models helped clarifying molecular, immunological, and genetic aspects that may play a role in the development of airway stenosis and in the healing process of the laryngeal-tracheal area. 

The aim of this narrative review is to present and discuss the state-of-the-art regarding LTS, focusing on its molecular mechanisms, mechanical and physio-pathological features, taking into consideration their clinical implications.

## 2. Molecular Basis of Abnormal Scarring of the Trachea and Subglottic Area

### 2.1. The Wound Healing Process in Human Trachea

The repair of a damaged soft tissue is a complex and tightly controlled molecular process. It has been demonstrated that the main steps of a repair process are conserved among mammals; notably, different organs’ repair processes share similar/overlapping features [4]. An organ-specific response to wounds may result in different healing outcomes; for example, whilst injuries in the dermis may heal by scarring, wounds in the oral mucosa mostly repair with minimal or no scar [5]. Wound healing occurs through a stereotyped response involving four programmed phases: haemostasis, inflammation, proliferation, and maturation/remodelling. These phases are often overlapping in space and time during a repair process. Any deviation from this typical sequence can lead to dysfunctional wound repair, excessive accumulation of matrix and ultimately tissue fibrosis. When this dysregulated repair process takes place in the airway tree, it may result in fibrotic stenosis. Haemostasis is triggered by the exposure of the subendothelial extracellular matrix (ECM) following tissue injury and blood vessels damage. It starts with the activation and aggregation of platelets through their adhesion to the ECM component via integrins and through the activation of G-protein coupled receptor (GPCR) pathways by soluble agonists secreted at the injury site [6]. The platelet-induced activation of the coagulation cascade results in the formation of a fibrin clot, that serves as a temporary ECM scaffold for the recruitment of circulating immune cells into the damaged tissue.

The inflammatory phase aims at the immune-mediated removal of pathogens and damaged cells before the formation of new tissue. Platelets’ α-granules induce inflammatory response both by expressing adhesion receptors, and by secreting a wide range of chemokines. Chemotaxis will then contribute to the recruitment of circulatory monocytes, neutrophils, and lymphocytes. Neutrophils are the first inflammatory cells recruited at the wound site. Their activation leads to the secretion of proteases (such as matrix metalloproteinases MMP_S_), and release of cytokines; the former will help in killing potential pathogens, whilst the latter will recruit monocytes and stimulate the proliferation of fibroblasts and epithelial cells [7]. Recent evidence suggests that platelets themselves may also play a role in the eradication of pathogens at the site of tissue injury. Activated platelets deliver effector molecules, such as kinocidins and platelet microbicidal proteins (PMP_S_), which directly attack bacterial, viral, fungal, and protozoan pathogens [8]. 

Neutrophils express MMP9, a major ECM-digesting enzyme, that degrades various intracellular matrix components, and activates the vascular endothelial growth factor (VEGF) to promote revascularization at the injury site. Neutrophil granules contain proteins enhancing the recruitment of monocytes These proteins, such as azurocidin (i.e., heparin binding protein) and cathepsin G, are released during neutrophils degranulation. After reaching the wound site at 48 to 72 h following injury, these cells phagocyte apoptotic neutrophils and dead cells [7].

The proliferative phase begins once monocytes differentiate into activated macrophages and shift their secretory pattern to different cytokines (IL-6, IL-1, fibroblast growth factor, epidermal growth factor, TGF-β, PDGF). At this stage fibroblasts, epithelial and endothelial cells proliferate and differentiate at the wound site. Angiogenesis promotes the development of a new capillary network resulting in the restoration of perfusion, thereby contributing to fibroblasts’ proliferation [9]. Activated fibroblasts differentiate into myofibroblasts, and express alpha-smooth muscle actin (α-SMA) that confers them contractile properties and capability to attach ECM components by extending their pseudopodia [10]. Myofibroblasts facilitate wound edges approximation through retraction of their pseudopodia; subsequent granulation and re-epithelization complete wound closure. The maturation phase can last from months to years with ongoing remodelling of the granulation tissue until the architecture of the permanent ECM is finalised. Metalloproteinases (MMPs) secreted by fibroblasts and resident inflammatory cells take part in transforming the ECM from its temporary form, made mostly by fibronectin and type III collagen, into its permanent structure composed primarily of type I collagen. As a result, the granulation tissue, which is highly vascularised and cellular is turned into an avascular and acellular scar. TGF-β signals and the mechanical load within the damaged tissue contribute to modulate the maturation phase [11]. Whenever synthesis and degradation of ECM become unbalanced, an excessive deposition of matrix components can disrupt the physiological architecture of the tissue or organ involved. In the airway tree this leads to an abnormal narrowing of the trachea, thereby favouring the onset of LTS. The healing process of the trachea and subglottic area have not been extensively studied, and therefore the pathophysiology and molecular mechanisms behind excessive scarring after an injury have not been fully clarified. As shown by recent research, a dysregulated response to mucosal injury can be caused by multiple factors interacting together. Genetic susceptibility coupled with growth factors, cytokines, altered fibroblasts as well as environmental factors (especially hypoxia and biomechanical stress) could all be implicated.

### 2.2. Fibroblasts Function in Laryngotracheal Stenosis

Fibroblasts are considered the key cells-effector in the deposition and reorientation of the extracellular matrix [12]. Fibroblasts are a highly heterogeneous cell population with different subtypes. Many studies analysing models of dermal injury suggest that scars are linked to a peculiar fibroblast cell lineage. Experimental studies identified a specific embryonic fibroblast lineage in murine dorsal skin responsible for the bulk of connective tissue deposition during cutaneous wound healing [13]. The reciprocal transplantation experiments by Rinkevich et al. showed that the differences in dermal architecture and would healing outcomes between oral versus dorsal dermis result from intrinsic fibroblast functions whilst are independent from the local microenvironment. In this scenario reticular fibroblasts of the dermis, Engrailed-1 positive, are activated by the inflammatory infiltrate in the dermis, and get involved in the development of excessive healing resulting in hypertrophic scar and keloid [14].

In recent years, several attempts were made to better characterise pathological fibroblasts implicated in chronic fibrosing diseases, in the attempt to find a unifying feature or pathophysiology behind them. Nuclear expression of c-JUN was documented in a fibroblast subset obtained from diverse fibrotic end-stage diseases, including idiopathic pulmonary fibrosis (IPF), scleroderma, myelofibrosis, kidney, pancreas and heart-fibrosis [15]. These experimental data also showed a decreased proliferation of fibroblasts after knockdown of c-JUN, as opposed to a rapid and widespread fibrosis after c-JUN induction. It was therefore suggested that c-JUN expression in fibroblasts could represent a common pro-fibrotic molecular mechanism across different human fibrotic conditions. The concept that aberrant healing in LTS may be related to the altered function of pathological fibroblasts has been tested in several studies.

A pioneering study by Macauley SP et al. analysed the steady-state of messenger RNA (mRNA) levels codifying for ECM proteins in cell cultures of human fibroblasts (taken from foetal skin, newborn foreskin and adult LTS). The variation of mRNA levels under the effect of TGF-Β-1 was also examined [16]. It was demonstrated that LTS fibroblasts reacted more sensitively to TGF-Β-1 compared with the other two cell cultures, as explained by the highest induction of mRNA levels for ECM proteins. Recent research highlighted that LTS-derived fibroblasts exhibit higher rates of migration and lower contractility compared to fibroblasts from normal airways. According to this study, the development and recurrence of LTS would be promoted by a specific fibroblast phenotype showing altered responsiveness to antifibroplastic signals during mucosal repair [17]. Prostaglandin E2 (PGE2) is an antifibroplastic molecule, modulating fibroblasts’ proliferation, migration, contraction, as well as synthesis and remodelling of ECM led by fibroblasts. In LTS, the significance of PDGE2 deficit during dysregulated wound healing is debated, even though data extrapolated from dermal fibrosis models suggest that insufficient PGE2 signalling is associated with aberrant fibroplasias [18]. Conversely, fibroblasts derived from human pathological specimens of LTS maintain normal PGE2 synthetic capabilities [19].

Data from rabbit models showed that LTS-derived fibroblasts demonstrated a lower response to PGE2 in terms of migration and contraction than fibroblasts derived from normal tracheal mucosa [17].

Comparing fibroblasts from scarred and uninjured areas, belonging to the same patients, it was found an imbalance between collagen production (collagen-1 and collagen-3 overexpression) and its breakdown (MMP2 downregulation) driven by pathologic fibroblasts [20]. LTS-derived fibroblasts also showed an aberrant metabolic profile with a shift towards glycolytic metabolism. Despite glycolysis being a less effective pathway in ATP generation compared to oxidative phosphorylation, it is more successful in conserving carbon atoms, which is advantageous for highly proliferative cells. Augmented glycolysis under aerobic conditions is an occurrence first described in cancer cells, known as “the Warburg effect” [21]. This metabolic anomaly seems to be also involved in the pathogenesis of IPF. Indeed, glycolytic cells reprogramming was described by Xie N et al. in IPF myofibroblasts, and its inhibition resulted in decreased fibrosis [22]. Some pathologic conditions such as diabetes may also affect fibroblasts phenotype and promote scarring in different types of tissues/organs.

Studies by Lina compared fibroblasts from LTS scar obtained from patients with and without type II diabetes mellitus. What they found in the first group was a myofibroblast phenotype, a metabolic behaviour characterized by increased oxidative phosphorylation and ATP production, and higher contractility [23,24]. This specific fibroblast phenotype may explain the higher risk (over eight-fold) of developing LTS in type II diabetes mellitus compared to no diabetes. It may also explain the clinical picture typical of this condition, characterized by high rate of posterior glottic stenosis with interarytenoid scar contraction limiting vocal fold mobility.

In summary, LTS displays a distinct fibroblast phenotype characterized by dysregulated response to antifibroplastic agents (i.e., PGE2) and by aberrant metabolic profile. These characteristics contribute to and are partly responsible for the healing outcome and the scarring recurrence in the trachea and subglottic area.

### 2.3. Role of Immune Dysregulation in the Pathogenesis of Iatrogenic LTS

Signalling between immune cells and fibroblasts is crucial for the development of fibrosis and scarring in LTS. Upper airway mucosal injury triggers an inflammatory response that results in upregulation of cytokines such as IL-1a, IL-6, IL-16, tumour necrosis factor a (TNF-a), fibroblast growth factor (FGF) and platelet-derived growth factor (PDGF). These cytokines then promote fibroblasts activation and differentiation into the myofibroblast phenotype [25]. Studies aiming at featuring the inflammatory infiltrate in LTS showed it mainly comprised T-cells, suggesting that a maladaptive T cell immune response may be involved in the pathogenesis of LTS [26,27]. The contribution of B and T cell immune response in the development of subglottic granulation tissue was well explained by Ghosh A et al. in a murine model of LTS [28]. In this experimental study, a mechanically or chemically injured laryngotracheal complex (LTC_S_) was implanted into the deep dorsal subcutaneous pockets of a wild type and severe combined immunodeficiency (SCID) mice. The tissue sections were examined at three weeks. The authors found that, despite significant granulation tissue formation was evident in wild type mice, direct airway injury had not induced the formation of granulation tissue under the damaged epithelium of airways’ mucosa in the SCID mice. This model suggests that adaptive cellular response and lymphocytes have a pivotal role in granulation and hypertrophic scarring after airway injury. More recent studies underlined the importance of CD4+ helper T-cells as the critical adaptive immune cells participating to the inflammatory response leading to tracheal stenosis. A prospective controlled study by Hillel AT et al. showed an increase in CD4+ T lymphocytic infiltrate and thickening of the lamina propria in human LTS specimens compared with normal controls (healthy laryngotracheal specimens) [29]. The timing of T lymphocyte infiltration after tracheal injury was assessed in a mouse model. At 4–7 days post injury a dense CD4+ T-cells infiltration was prevalent, whilst on 14–21 days fibrosis was developing. To support a role of TH2 immunophenotype in the pathogenesis of LTS, the same authors demonstrated a higher IL4 protein and gene expression in iatrogenic specimens, compared to controls. The involvement of TH2 cells in orchestrating the fibrous tissue microenvironment has been described in different fibrosing diseases. Type 2 immunity suppresses type 1- and TH-17-driven inflammation, and induces tissue repair and regeneration after injury. However, the tissue regenerative type 2 response may result in progressive fibrotic disorders, especially in cases of persistent activation of tissue repair pathways [30]. Several experimental models have been developed to investigate the molecular mechanisms of wound healing and fibrosis. They have shown that TH2 response involves IL-4, IL-5 and IL-13 and promotes collagen deposition and ECM remodelling, whereas TH1 responses, together with the production of interferon-γ (IFN-γ), exhibits antifibrotic action [31]. The imbalance between TH1 and TH2 responses may therefore drive the development of tissue fibrosis in different organs. Chronic helminth infections, such as severe schistosomiasis, are associated with fibroproliferative lesions elicited by type 2 immune responses [32]. Type 2 cytokines (IL-4, IL-13) and their receptors are increased in patients with IPF [33].

Furthermore, the cytokine IL-13, activates macrophages and promotes their polarisation to a profibrotic M2 phenotype, which seems to stimulate fibrosis in different tissues [34]. Based on protein and gene expression, macrophages are classified in classically activated macrophages (M1), which are induced by TH1 cytokines (IFN-γ) and by infections, and alternatively activated macrophages (M2), that are induced by TH2 cytokines such as IL-4 and IL-13. During repair, at the early stage of tissue injury, M1 response acts through the secretion of proinflammatory cytokines and anti-microbial mediators, killing microorganisms and contributing to ECM deposition via fibroblasts activation. Later on, in the resolution phase, M1 macrophages differentiate into M2 macrophages, stimulated by microenvironment changes, (efferocytosis, PGE2, IL-1R and others). M2 macrophages are thought to act in the late stage of tissue injury exhibiting anti-inflammatory activity and promoting wound healing and tissue remodelling. However, evidence on the topic is conflicting, as M2 macrophages are also considered a phenotype promoting pathologic deposition of extracellular matrix leading to tissue fibrosis, especially in the setting of persistent M2-mediated inflammation. This would be executed through the expression of profibrotic factors such as TGF-Β1 and arginase-1 [35]. Using an “in-situ in-vivo” bleomycin induced mouse LTS model, Hillel AT et al. showed a rise in M2 phenotype after 10 days after laryngo-tracheal injury, suggesting that dysregulated M2 macrophages play a role in abnormal tracheal wound healing [36]. Recently, Motz K et al. described an increased number of M2 macrophages in an immunohistochemical analysis of human laryngotracheal scar specimens. In the same study, they also showed that in vitro M2 macrophages favoured aberrant fibroblasts behaviour and promoted tracheal fibroblasts to express and produce collagen [37]. These findings are consistent with similar investigations on other fibrotic diseases which underline the potential role of M2 inflammation in aberrant ECM deposition, especially in IPF and cardiac fibrosis [38,39].

### 2.4. Immune Response in the Pathogenesis of Idiopathic Subglottic Stenosis

Idiopathic subglottic stenosis (iSGS) is a rare clinical entity occurring in a strikingly homogeneous population (female, Caucasian, perimenopausal). Therefore, it may not be surprising that a slightly different immunological derangement than iatrogenic LTS may be at the basis of the inflammatory-fibrotic response that results in a short, concentric stenosis at the cricoid cartilage level. Recent data comparing human iSGS to healthy control specimens have begun to clarify the molecular pathogenesis underlying the fibrosing phenotype of iSGS. Gelbard A et al., analysed 20 human iSGS and were able to confirm a pronounced ECM component in the subglottic scar area; this was associated with an up-regulation of the IL-17A/IL23 inflammatory axis within infiltrating immune cells (γδT-cells) [26]. The same research group conducted an experimental in vitro study on primary fibroblast cell lines from iSGS subjects that showed an increased fibroblast proliferation and amplified inflammatory response under IL-17A stimulation, and realized through myeloid-recruiting chemokines (GM-CSF and CCL2) and cytokines (IL-6) [27]. The results of this study also described a synergistic action between IL-17A and TGF-Β1 in supporting ECM deposition and subglottic fibrosis. Interestingly, iSGS disease preponderance in female may relates to a hormonal role on host inflammation and fibrosis. In the experimental study by Morrison RJ et al., estradiol did not show any link with fibrosis development. Other recent studies speculated that laryngotracheal microbiota may also provide an essential contribution in promoting iSGS. A disrupted microbiota could locally activate γδ T cells, which participate in the early immune response to a range of infectious agents at mucosal sites and main source of IL-17A [40]. In a recent study, Hillel AT et al. identified the aetiology of the microbiota in iSGS, iatrogenic LTS and patients without LTS. iSGS seems to be associated with the *Moraxellaceae* family, which includes the genera *Moraxella* and *Acinetobacter*. It was also described an inverse correlation between *Prevotella* and *Streptococcus* among all samples. This finding is of particular interest, as *Prevotella* are commensal bacteria in the airways, which acts preventing pathogens’ adherence at mucosal sites. In iSGS, the loss of *Prevotella* may lead to pathogenic bacterial proliferation, dysbiosis and ending up in scarring and fibrosis [41].

A study by Gelbard A. et al., analysing a series of human LTS specimens, demonstrated that *Mycobacterium* species are uniquely associated with iSGS. This finding added evidence to the hypothesis that altered local microbial flora coupled with a dysregulated inflammatory host response could be partly responsible of laryngo-tracheal fibrotic processes [42]. Figure 1 shows the interaction between immunological response and tracheal scarring in iatrogenic and idiopathic tracheal stenosis.

### 2.5. TGF-Β in the Development of LTS

TGF-Ββ is considered one of the main growth factors driving fibrosis in different organs and tissues. Three prototypic TGF-Β isoforms, TGF-Β1, TGF-Β2 and TGDF-β3, intervene in regulating early embryonic development, in maintaining tissue homeostasis, in wound-healing responses.

TGF-Β is secreted by a number of cell types, as an inactive precursor bound to latency-associated peptides, which is subsequently activated in the ECM by proteases cleavage (elastase, MMPs) [11]. Active TGF-Β exerts its biologic effects on target cells by binding to a transmembrane receptor with a strong serine/threonine kinase activity and a weak tyrosine kinase activity. The subsequent activation of the TGF-Β downstream pathway involves canonical signalling, with the activation of cytoplasmic proteins known as the SMADs (small mothers against decapentaplegic), but also noncanonical signalling. Both canonical and noncanonical signalling take part in the epithelial-to-mesenchymal transition (EMT), a biological process entailing a functional transition of polarized epithelial cells into mobile mesenchymal cells able to secrete ECM component. This pathway may be activated in response to tissue repair and to pathological stress. Cytoskeleton rearrangements and tight junctions dissolution are facilitated by noncanonical signalling, and translate in enhanced cells migratory capacity and invasiveness. SMAD-mediated pathways (i.e., canonical signalling) result in activation of different target genes such as collagen, plasminogen activator inhibitor-1 connective tissue growth factor, and α-SMA, and in inhibition of some epithelial genes, promoting epithelial-mesenchymal transformation [43]. The EMT process mediated by TGF-Β action is involved in fibrosis occurring in different organs such as kidney, liver, lung and intestine. TGF-Β also plays a key role in the differentiation and activation of myofibroblasts, inducing the expression of α-SMA through canonical signalling, via SMAD-binding elements, which positively regulate ACTA2 transcription (encoding α-SMA). In pathological conditions, myofibroblasts may evade apoptosis by activating molecular mechanisms in response to pro-survival biomechanical and growth factor signals derived from the fibrotic microenvironment, leading to excessive synthesis deposition and remodelling of ECM, which finally results in exuberant tissue scarring. Experimental studies analysing IPF and systemic sclerosis fibroblasts showed that persistence of myofibroblasts and resistance of apoptosis are TGF-Β1 mediated mechanisms via the activation of the focal adhesion kinase (FAK)-protein kinase B (PKB also known as AKT) signalling pathway, which results in phosphorylation and inhibition of the sensitizer protein B-cell lymphoma 2 (BCL-2)-associated death promoter (BAD) [44]. BAD indirectly promotes apoptosis through intrinsic pathway, binding and blocking pro-survival proteins such as BCL-2, BCL-X_L_, BCL-W, which induce mitochondrial outer membrane permeabilization. A study by Scioscia KA et al. in biopsy specimens obtained from patients with subglottic stenosis demonstrated an upregulation of TGF-Β compared to healthy controls suggesting a role of TGF-Β in the pathogenesis of LTS [45]. Data extrapolated from animal models showed that hyperexpression of TGF-Β appears early in the course of laryngeal wound healing, and returns to baseline levels within 21 days. Moreover, in murine models of subglottic stenosis, the infusion of neutralizing antibodies versus TGF-Β demonstrated to reduce the expression of ECM proteins and scarring process [46]. In a subsequent study of induced subglottic airway injury performed in a canine model, a combination of intralesional and intravenous anti-TGF-Β was able to determine a significant reduction in tracheal stenosis and an increase in survival time, compared to the saline control subject [47]. Overall data obtained from experimental studies show the key role of TGF-Β in the pathophysiology of laryngotracheal aberrant healing process, and suggest a potential therapeutic role in the modulation of TGF-Β pathways.

### 2.6. Role of Hypoxia as Promoter of Laryngotracheal Scarring 

Microenvironment changes induced by biochemical (hypoxia) or biomechanical stimuli may act on local mucosa promoting collagen deposition and tissue stiffness, contributing to aberrant tracheal healing. Endotracheal intubation may result in tracheal ischemia especially in critically ill patients requiring prolonged mechanical ventilation [48]. The segmental tracheal blood supply and the narrow anatomy of the subglottic area facilitate ischemia and local necrosis of the mucosa following the mechanical compression exerted by the tube and the cuff, resulting in hypoxic cellular injury. Despite hypoxic microenvironment in a damaged tissue activates compensatory mechanisms which allow the tissue to adapt to low oxygen, such as angiogenesis, prolonged hypoxia may stimulate pathological repair and fibrosis. Hypoxia-inducible factor-1 (HIF-1) is a transcription factor which carries out the main physiological response to hypoxia. HIF-1 is composed of oxygen sensitive HIF-1α and constitutively active HIF-1β subunits. In a tissue environment characterized by sufficient oxygen, HIF-1α is hydroxylate and subjected to proteasomal degradation [49]. During hypoxia, HIF-1α heterodimerizes with HIF-1α, provoking cellular metabolic reprogramming via activation of target genes involved in hypoxic adaptation. HIF-1α mediated glycolytic reprogramming is necessary for myofibroblast differentiation and fibrotic progression as shown by a study about pulmonary fibrosis on animal model [50]. Furthermore, HIF-1α proved to have a key role in upregulation of different profibrotic genes, including plated derived growth factors (PDGFs), fibronectin and connective tissue growth factors, which in turn are able to promote collagen deposition in different organs [51,52,53]. Hypoxia may exert his fibrotic effect also by inducting profibrotic cytokines such as IL-6, which in turn increases the expression of α-SMA, collagen-1 and MMP13 in normal laryngotracheal fibroblasts culture, helping them shifting into the myofibroblast phenotype in LTS [54]. These experimental data suggest that the hypoxic microenvironment facilitated by prolonged intubation may play a role in the development of aberrant laryngotracheal healing. 

### 2.7. Genetic Susceptibility to Aberrant Tracheal Healing 

Even though tracheal and laryngeal injuries occur in more than half of patients who receive mechanical ventilation, only a small subgroup of intubated patients develop a clinically significant LTS [55]. Abnormal wound healing responses partly depend on the genetic background of affected individuals. A case–control study analysing single nucleotide polymorphism (SNPs) focused on TGF-Β and described the association between a functional SNP of TGF-Β1 (−509 C/T) with the development of LTS after tracheal intubation. A genetic predisposition could therefore contribute to the pathogenesis of abnormal laryngotracheal scarring [56]. This study suggested that TGF-Β1 SNP −509 C/C rs1800469 is associated with increased susceptibility for stenosis, while genotype −509 C/T rs1800469 may provide a protective function against LTS development. Another, more recent pilot case–control study, recruiting forty patients with LTS and thirty-six control patients with airway injury but without LTS, identified an association between tracheal exuberant scarring and genetic polymorphism MMP-1 rs1799750. This SNP of MMP-1 is able to enhance the expression of MMP-1 induced by exposure to mechanical forces [57]. A polymorphism in CD14 rs2569190 resulted protective against LTS development, presumably due to lower susceptibility to bacterial infection and altered local microbial flora after tracheal injury. CD14 is an innate immune receptor involved in innate defence against infections. It binds to bacterial lipopolysaccharides and contributes to recognise lipoproteins of mycobacterium tuberculosis. A subsequent case–control study examined the association of candidate wound healing genes polymorphisms (MMP1, MMP3, MMP12, CD14, TGF-Β and MCP1) with the presence or absence of LTS. Although no significant association between SNPs and LTS development was found in the overall population, a subgroup analysis revealed that SNPs are associated with LTS depending upon ethnic background. In particular, the AA genotype of MMP12 rs2276109 was more prevalent among African Americans with LTS than Caucasian and Hispanics with LTS, emphasizing the importance of different ethnic backgrounds [58].

### 2.8. Programmed Cell Death Protein 1 Pathway in LTS

Programmed cell death protein 1 (PD-1) receptor (also known as PDCD1 and CD279), through the interaction with its ligands PD-L1 (B7-H1; CD274) and PD-L2 (B7-DC; CD273), plays a critical role in maintaining immune homeostasis, serving as immune checkpoint on effector T cells [59]. Particularly, PD-1 pathway activation exerts an essential inhibitory function on T-cell in the setting of chronic infection and cancer, where persistent antigen stimulation can lead to T cell exhaustion. Although PD-1 pathway has received considerable attention for its role in T effector cells inhibition and cancer immunosuppression, PD-1 is not an exhaustion-specific marker, as demonstrated by its expression also in cells such as regulatory T cells (Tregs), B cells, natural killer cells and some myeloid cells. However, beyond the complex and wide-ranging functions of PD1 on immune regulation and tolerance, recent data showed that PD-1/PD-L1 axis may be also implicated in the pathogenesis of fibrotic and granulomatous diseases. Upregulation of PD-1 and PD-L1 has been described in interstitial lung diseases where blockade of the PD-1 pathway has proven to be capable of restoring T-cell function and proliferative capacity in granulomatous diseases, such as chronic beryllium disease and sarcoidosis [60]. Furthermore, PD-1 is overexpressed on CD4+ T cells obtained from blood and lung of patients affected by IPF. Moreover, experimental data showed that in pulmonary fibrosis PD-1 up-regulation on human TH 17 cells promote a profibrotic microenvironment through increased secretion of IL-17A and TGF-Β via the STAT3 signalling pathway. Blockade in PD1/PD-L1 axis in CD4+ cells allows to reduce IL-17A expression, resulting in inhibition of collagen-1 production [61]. In a murine model of IPF, RNA sequencing analysis found an upregulation of PD-L1 in lung fibroblasts which exhibit an invasive phenotype, while PD-L1 inhibition resulted in a significant attenuation of lung fibrosis [62]. These data suggest a role of PD-1 pathway in promoting fibrotic microenvironment and dysregulation of ECM composition and stiffness which may contribute to several pathological conditions such as pulmonary fibrosis but also LTS. A recent controlled ex vivo study by Davis RJ et al., using immunohistochemical staining of laryngotracheal resection, showed a higher expression of both PD-1 and PD-L1 in tissue obtained from iatrogenic LTS, and higher PD-1 expression in patients with iSGS, compared to control [63].

Furthermore, the observation of spatial association of PD-1, PD-L1 and CD4 expression in tissue specimens suggests that PD-1/PD-L1 interaction could occur on CD4 T cells promoting fibrotic change of laryngotracheal tissue [64]. In the same study, fibroblasts isolated from scars of patients suffering from iatrogenic LTS showed a significant overexpression of PD-L1, collagen-1 and fibronectin-1, after TGFβ1 stimulation, compared to untreated scar fibroblasts. Therefore, in the laryngotracheal scar microenvironment, TGF-Β1 may act promoting fibroblasts overexpression of PD-L1, supporting prior findings reported on pulmonary fibroblasts [65].

## 3. Physical Stimuli and Mechanotransduction in the Pathogenesis of Tracheal Stenosis

### 3.1. Mechanical Behaviour of the Trachea 

The trachea is a unique part of the conducting airways consisting of the following three main components responsible for its mechanical behaviour: cartilage rings, smooth muscle and connective tissue. Cartilaginous rings maintain the trachea open during the respiratory movement induced by intrathoracic pressure swings and play a major role in the mechanical function of the trachea. Tracheal cartilage, like other hyaline cartilages, is a fibre-reinforced composite in which a proteoglycans matrix core is trapped by collagen fibres arranged in an interlacing network of fine fibrils. Tracheal cartilage exhibits the properties of a nonlinear material, despite being widely treated as a linear elastic material in different studies [66]. Spontaneous breathing and mechanical ventilation exert physical forces to the tracheal wall and its components. According to the principle of mechanotransduction, abnormal physical stimuli may trigger cellular response through the activation of specific genetic programs [67]. In line with this, pathological changes secondary to tracheal stenosis may act as a cellular trigger, promoting further scarring or contributing to scar recurrence. The mechanical behaviour of a tissue subjected to forces or motion producing different forms of loading is described by stress (σ) and strain (ε) relationship. Stress is defined by the ratio of the force (F) to the cross-sectional area (A) of the tissue:$$\sigma =\frac{F}{A}$$

Strain defines the deformation of the material resulting from the application of stress:$$\varepsilon =\frac{L-L_{0}}{L_{0}}$$
where ε is strain, L is length after load is applied, and L_0_ is equivalent to original length. Linear elastic material behaviour and the stress–strain relationship is defined by Hooke’s law:σ=E · ε
$$E=\frac{\sigma }{\varepsilon }$$
where E is the modulus of elasticity (Young’s modulus) which represents a measure of stiffness dependent on the nature of the material itself. Studies analysing stress–strain relations obtained from slices of human tracheal cartilage showed a linear behaviour for deformations up to 10% of the initial length of the samples. For deformations exceeding this limit, strain hysteresis and residual strain increased progressively [68]. In physiological conditions, the trachea is subjected to a strain not exceeding 5%; thus, the mechanical properties of tracheal tissue follow the Hooke’s law. Studies measuring Young’s modulus through tensile tests on tracheal cartilage, reported a value ranging from 1.8 to 15 MPa, depending on the age of the patient [69]. However, the mechanical properties of tracheal cartilage can be altered in case of pathological processes. Among these, tracheomalacia (TM) represents a condition of excessive tracheal collapse attributed to reduced cartilage stiffness and hypotonia of the myo-elastic elements. This disorder may be congenital (primary TM), or secondary (acquired TM), like in complex tracheal stenosis where the stenosis exceeds 1 cm and involves cartilage. The main physiological consequence of the mechanical derangement of tracheal cartilage is a change in the stress/strain relationship as shown in different animal models. Softening tracheal cartilage by papain intravenous injection in rabbit results in a significant reduction in maximal expiratory flow, while enzymatic digestion of structural components of the trachea causes an expiratory flow limitation associated with tracheal collapse [70,71]. In a computational non-linear finite element model, Hollister S.J. et al., showed that a critical reduction in non-linear properties (even limited to a 1 cm segment) in cartilage rings, caused a sudden mechanical airways instability and tracheal collapse; conversely, the reduction in posterior tracheal smooth muscle properties was able to decrease exhalation area without tracheal collapse [72].

During spontaneous breathing the trachea in subjected to stress and strain, where stress is defined by transmural pressure through the tracheal wall, whilst strain is defined by tracheal wall deformation. These forces are strictly linked to the presence of airway stenosis and its physiological changes, but they are also dependent on the mechanical properties of the tracheal wall.

### 3.2. Physiological Changes in LTS 

LTS increases airway resistance and decreases airflow during breathing. Resistance to airflow is described by the Hagen–Poiseuille equation, which defines the relationship between airway radius and resistance.
$$R=\frac{8µL}{{\pi r}^{4}}$$
where *R* is resistance to airflow, µ is viscosity, l is length, and r is airway radius.

Based on this assumption, a 50% tracheal radius reduction (e.g., stenosis) results in a 16-fold increase in airflow resistance. Although helpful in understanding the physiology of LTS, this equation only applies to fully developed laminar flow in straight circular tubes. The Hagen–Poiseuille equation is, however, unreliable in the upper respiratory tract, where airflow nearly changes from laminar to turbulent and the anatomy of the airways has turns and bends. The Bernoulli obstruction theory better describes upper airway resistance. It states that airway resistance is inversely proportional to airspace cross-sectional area and:$$R\propto A^{-1}$$

Following this principle, a four-fold reduction in cross sectional area would result in a four-fold increase in upper airway resistance [73]. The Venturi effect describes the relationship between airflow velocity and pressure variation through a tube with a stricture, representing a physical model for tracheal stenosis:$$P+\frac{1}{2}\rho v^{2}=constant$$
where *P* is the pressure in the tube and *v* is airflow velocity. As per equation, a decrease in the section of the tube-trachea corresponds to an increase in the velocity of the airflow, and a decrease in pressure to keep the total energy contained within the system constant. A pressure drop in the upper airways acts as physical force directed inwardly on the regional tracheal wall at the level of stenosis. This would change the stress–strain level during spontaneous breathing, possibly reaching the non-linear stress–strain relationship threshold (Figure 2). Furthermore, especially in stenoses complicated by TM, the negative pressure generated through spontaneous breathing causes an inward mural collapse during inspiration, worsening the aforementioned physiological effects.

The pressure fall (Δ*P*) over the airway stricture is the main determinant of the patients’ work of breathing, even though it becomes significant only when the stenosis is severe. In a computational fluid dynamic study on a realistic model of tracheal stenosis, the simulated δ*P* over the stenosis was seen to dramatically increase only when more than 70% of the tracheal lumen was obliterated. However, the same study showed that δ*P* is flow dependent and increases proportionally with the increase in the airflow as described by the equation below:$$\Delta P_{\left[stenosis\right]}^{Loss}=\rho K\frac{Q^{2}}{2S^{2}}$$
where Q is bulk flow, *S* is local cross section, ρ is gas density (1.2 kg/m^3^ for air), and K is an empirically determined constant for any specific geometrical feature (being estimated = 1.2). This physical concept well explains why the onset of symptoms during spontaneous breathing occurs at an advanced stage of tracheal stenosis. However, considering that the fall in pressure is flow dependent, the onset of symptoms may also occur in moderate stricture in conditions of increased airway flow during breathing (e.g., physical exertion) [74]. 

### 3.3. Respiratory Drive in LTS

Upper airway obstruction is often associated with a hyperactivation of the respiratory drive and the onset of an excessive negative intrathoracic pressure to overcome flow limitation due to stenosis. In patients suffering from LTS, the generation of more negative intrathoracic pressure to overcome upper airway airflow reduction results in an additive effect on upper airway resistance due to the Bernoulli effect described by the equation:$$R_{STENOSIS}=\frac{1}{C_{D}}\;\sqrt{\frac{\rho \Delta P}{2}}\;\frac{1}{A}$$
where *C_D_* is the discharged coefficient (function of the Reynold number and the geometry), Δ*P* is the pressure drop over the stenosis, ρ is gas density (1.2 kg/m^3^ for air), and A is the airspace cross-sectional area at the constriction. The anatomical site of stenosis is another important factor influencing airflow dynamic; specifically, glottic and subglottic stenosis affect resistance considerably more than distal trachea strictures [75].

ΔPes (Delta esophageal pressure) is a reliable measure of pleural pressure changes during breathing, but studies analyzing esophageal pressure swings (ΔPes) are lacking. Some data can be extrapolated by physiological measurements obtained from the paediatric population. Argent A.C. et al., analysed δPes in children with acute severe airway obstruction due to viral croup. ΔPes measured values were five times higher in the croup patients than in the control, with median peak-to-trough oesophageal pressure changes of 58 cm H_2_O (range 27 to 120 cm H_2_O) in croup patients vs. 11 cm H_2_O (range 7 to 12.5 cm H_2_O) in controls [76]. These ΔPes values are much higher than those recorded in adult patients affected by acute de novo respiratory failure [77]. Moreover, patients with severe croup showed a significant lower dynamic compliance than controls, suggesting an increase in lung weight, presumably due to negative pressure pulmonary edema. The rapid onset of pulmonary edema after inspiratory effort against an obstructed airway was reported since 1973 and associated with different causes of upper airway obstruction such as laryngospasm, bronchial occlusion, foreign body aspiration and upper airway tumour. This post-obstructive pulmonary edema is secondary to an excessive negative pleural pressure swing due to the magnitude of the inspiratory effort which is transmitted to lung interstitium and alveolar spaces, resulting in increased trans-vascular fluid filtration and hydrostatic edema formation (i.e., negative pressure pulmonary edema) [78]. Thus, a breathing pattern characterized by vigorous inspiratory effort in patients suffering from severe LTS could potentially result in negative pressure pulmonary edema with the onset of acute respiratory failure. Excessive inspiratory effort also increases inspiratory flow and pressure drop in the site of stenosis. This process may promote the activation of mechanotransduction pathways through pathological physical stimuli on the tracheal wall (Figure 2).

### 3.4. Mechanotransduction in Tracheal Scarring

Despite studies regarding the contribution of mechanical stimuli on dysregulated wound repair in laryngotracheal tissue are lacking, data from studies conducted in other tissues which clarified the interplay of mechanical-stress strain and excessive scarring suggest a potential role of biomechanical pathways in LTS pathogenesis. 

Large organisms have evolved to compensate the physical forces (i.e., stress and strain) acting on tissues and organs through different mechanisms, among which mechanotransduction, in order to maintain organ mechanical properties. This adaptation, in human and other large organisms, may promote fibrotic response and exuberant scarring during tissue healing; in contrast, in smaller model organisms, who are subjected to lower physical stress, healing occurs by regenerating normal tissue architecture without fibrosis [79,80]. 

Skin sites subjected to unphysiological mechanical stress exhibit a high propensity for keloid formation [81]. Keloids usually occur at sites that are frequently subjected to mechanical forces (i.e., anterior chest, scapular regions), while they rarely occur in areas of low stretching/contraction (i.e., anterior lower leg). However, in this complex process, the activation of mechanotransduction pathways at sub-cellular level in fibroblasts may play a critical role [82]. Physical forces act on biological systems through ECM proteins via integrins interaction at the cell’s surface, allowing the transmission of external mechanical stimuli to cells, influencing actin cytoskeleton’s tension, integrin-mediated focal adhesion, and triggering downstream cellular signals. Focal adhesion kinase (FAK) is a well-characterized transducer of physical forces captured by ECM proteins-integrins machinery, which is able to initiate downstream intracellular pathways involved in the activation of different profibrotic genes. This process may contribute to fibrosis in different organs subject to unphysiological mechanical stimuli, including the lung [67,83]. Experimental data showed that tissue exposure to chronic mechanical stress may promote persistent myofibroblast activation and ECM proteins production, resulting in matrix stiffening, which in turn acts amplifying progressive fibrosis via suppression of fibroblasts expression of cyclooxygenase-2 and PGE-2 (an autocrine inhibitor of fibrogenesis) [84]. In a large animal porcine model, blocking mechanotransduction through disruption of the FAK pathway results in improvement in scar formation, enhancement of skin regeneration, and promotion of a regenerative fibroblasts phenotype reducing fibrosis [85]. Due to the fact that physiological changes described in patients suffering from LTS are presumably a source of unphysiological mechanical stress on laryngotracheal tissue, it is likely that the resulting change of local micro-environment may contribute to dysregulated wound repair through mechanotransduction pathways. In particular, in spontaneously breathing patients suffering from LTS, shear stress may promote worsening of stenosis via local mechanotransduction, as air flow increases in conjunction with a reduction in airway diameter. This biophysical phenomenon may also be at the basis of the frequent tracheal scarring relapses after endoscopic surgery (Figure 2).

## 4. Clinical Implications

Although topical mitomycin-C application and intralesional steroid injection have been widely used for re-stenosis prevention after endoscopic surgery, clinical benefits of these treatments are minimal and are unlikely to significantly change the natural evolution of the disease. Pharmacological approaches based on antifibrotic medications are currently being tested at pre-clinical level. These treatments may have the potential to open up new prospects for future treatments aimed at regulating immunological derangement and/or slowing down the deposition of pathologic ECM in LTS. Antifibrotic medications currently approved for the treatment of IPF (i.e., pirfenidone and nintedanib) have been used in animal models of LTS. Pirfenidone exerts antifibrotic properties through the suppression of TGF-Β1 activity. Pirfenidone also exhibits anti-inflammatory effects via cytokine downregulation, inhibition of inflammatory cells accumulation, and limitation of the oxidative stress response [86]. In an experimental model of tracheal wound healing Pirfenidone decreases inflammation, collagen deposition and TGF-Β1 expression [87]. In a rat model, intraperitoneal application of pirfenidone for 10 days reduced fibrosis and narrowing of the intratracheal lumen diameter, compared to the control group in which 1 mL of saline solution was injected [88]. Nintedanib is a triple tyrosine kinase inhibitor and growth factor antagonist. It targets the receptors of VEGF, PDGF and FGF and can interfere with both classic and non-classic TGF-Β1 signalling pathways. In a preclinical rat model, nintedanib proved to effectively prevent tracheal stenosis by inhibiting fibroblast proliferation, migration and differentiation, and by suppressing TGF-Β1/SMAD2/3 and ERK1/2 signalling pathways [89]. As suggested by preclinical data, there is a rationale for using drugs that specifically block TGF-Β pathway in LTS. A number of anti-TGF-Β inhibitors and humanized monoclonal antibodies of recent development seem to be promising in preclinical studies. At present however clinical translation of the pre-clinical data obtained with TGF-Β activity suppression is affected by significant concerns regarding efficacy and safety [90]. This might be related to the wide range of pleiotropic biological functions of TGF-Β.

Rapamycin is a macrocyclic antibiotic with immunosuppressive and antiproliferative functions used in solid organ transplants. It acts as an inhibitor of the mammalian target of rapamycin (mTOR). The capacity of mTOR to modulate ECM production in fibroblasts, made of it a novel therapeutic target potentially able to impact the pathophysiology of different fibrotic diseases [91]. Results of an in vitro study showed that rapamycin has anti-fibroblastic effects on LTS, inducing a decrease in LTS fibroblasts proliferation, function and metabolism [92]. Drug-eluting stents could be a way of administering drugs locally, directly to the trachea avoiding systemic side effects. In a murine model of LTS, a rapamycin-eluting stent reduced the thickness of the lamina propria and the expression of collagen 3, TGF-Β and α-SMA. This prospects a potential role of the rapamycin-eluting stent in preventing tracheal stenosis recurrence after endoscopic surgery [93]. Tacrolimus is an FDA approved immunosuppressive agent used as anti-rejection agent in recipients of solid organs transplant. It exerts its actions through calcineurin inhibition preventing activation of the nuclear factor of activated T cell (NFATc). NFATc is a transcription factor activated by dephosphorylation via the calcineurin pathway, which induces upregulation of IL-2 causing T-cell growth and differentiation. Calcineurin inhibition induced by tacrolimus causes the suppression of T-cell activation. In an acute tracheal injury rat model, a low dose of systemic tacrolimus was effective in inhibiting the activation of immune cells in the airway mucosa during the early stage of wound healing, and in the prevention of LTS [94]. Furthermore, a study recruiting 19 patients affected by benign airway obstruction treated with the use of self-expanding metallic airway stents suggested that an immunosuppressive tacrolimus containing regimen may be able to prevent granulation tissue formation and airway narrowing [95]. Other potential targets for LTS treatment include IL-17 for iSGS, and PD-1/PD-L1 pathways, although no preclinical data are currently available. 

Table 1 summarizes the available pre-clinical studies potential pharmacological treatment of fibrotic LTS.

## 5. Conclusions

LTS is a multifactorial and heterogeneous disease characterized by excessive fibrotic response to airway injury. Recent evidence shows that this fibroinflammatory process is the result of an immunological dysregulation leading to fibroblasts activation and differentiation into myofibroblasts. Although the complex interplay between physical stimuli and biological response in LTS still needs to be clarified, data obtained from studies that analysed other tissues suggest that mechano-transduction is implicated in the worsening of tracheal scarring and LTS relapse after endoscopic surgery. Therefore, the alteration of the macro and microenvironment in the site of airway injury due to pathological stress–strain on tracheal tissue but also due to local hypoxic stimuli may promote fibrotic response. At present the therapeutic strategies to treat LTS include laryngotracheal resection and reconstruction, even though endoscopic treatment is spreading as a less invasive treatment option. However, endoscopic but also open surgery are burdened by relapses requiring multiple treatments, and tracheostomy in some cases, with a significant negative psychosocial impact and reduction in quality of life. Overall, the development of non-surgical treatments based on disease physiopathology is needed to prevent excessive tracheal scarring after prolonged endotracheal intubation and LTS recurrence after endoscopic and surgical treatment. Although the knowledge of the biological processes behind LTS development is still limited, some biological targets are suggested by preclinical studies and open up new perspectives for future medical treatment.

## Figures and Tables

**Figure 1 ijms-23-02421-f001:**
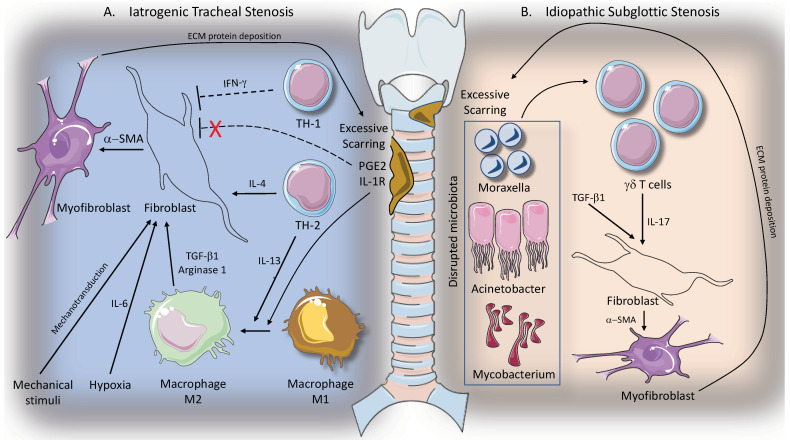
Immunological response and tracheal scarring in iatrogenic and idiopathic tracheal; Panel (**A**): iatrogenic tracheal stenosis is sustained by lymphocyte TH2 response that promotes fibroblasts activation and myofibroblasts differentiation. Furthermore, TH2 response through IL-13 induction promotes alternative polarization of macrophages to a profibrotic M2 phenotype, which in turn acts by inducing fibroblast activation via TGF-Β1 and arginase-1 expression. See text for more details; Panel (**B**): idiopathic subglottic stenosis seems related to γδT cell activation as a consequence of disrupted tracheal microbiota. γδT cells induce activation of the IL-17A/IL23 inflammatory axis, which acts in synergy with TGF-Β1 in promoting myofibroblasts differentiation and scarring development.

**Figure 2 ijms-23-02421-f002:**
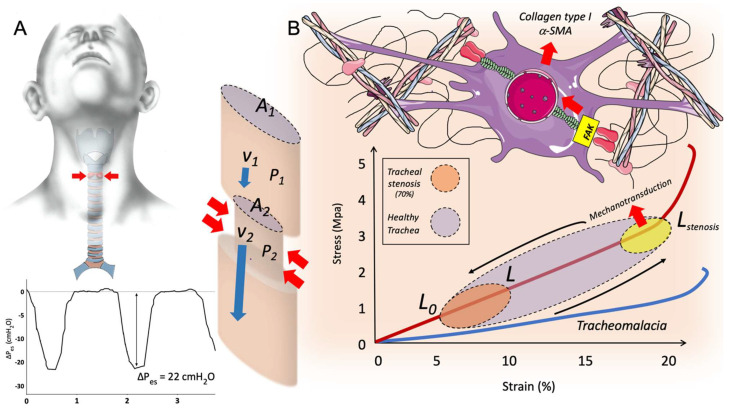
Physiological changes in LTS stenosis; Panel *(***A***):* In normal tracheal section (A1), physiological airways pressure (P1) is associated with normal airflow velocity (V1). In patients suffering from LTS, a hyperactivation of the respiratory drive occurs to overcome airways resistance at the site of stenosis. The inspiratory effort determines high inspiratory flow and elevated pleural pressure swing, as illustrated by ΔPes. At the site of stenosis (A2), according to the Venturi effect, the increase in airflow velocity at stenosis level (V_2_) is associated with a reduction in airway pressure (P_2_) resulting in a pressure drop, which acts on the tracheal wall. These physiological changes may exert unphysiological stress and strain on the tracheal mucosa at the site of stenosis during breathing in patients with LTS; Panel (**B**): Stress–strain relationship in healthy trachea and in tracheomalacia. During breathing, tracheal wall is subject to mild mechanical stimuli, which results in a negligible tracheal deformation (strain < 10%). In patients with severe LTS (reduction in patency > 70%) pressure drop at the site of stenosis increases mechanical stimuli on the tracheal wall resulting in unphysiological stress–strain. The consequent changes of mechanical microenvironment may activate mechanotransduction pathways in local fibroblasts, promoting myofibroblast differentiation and collagen deposition; *V_1_: airflow velocity before stricture, V_2_: airflow velocity at stenosis site, P_1_: pressure in the airway before stricture, P_2_: pressure in the airway at stenosis site, A_1_: area of healthy trachea, A_2_: area at stenosis sit*.

**Table 1 ijms-23-02421-t001:** Pre-clinical studies on LTS treatments.

Study	Treatment	Mechanisms of Action	Study Model	Outcome
Olmos-Zuniga et al., 2017 [87]	Pirfenidone	Suppression of TGF-b1 Downregulation of cytokine production Inhibition of fibroblast proliferation	Animal model 40 rats undergoing cervical tracheoplasty Group 1: saline solution Group 2: collagen-polyvinylpyrrolidone Group 3: mitomycin C Group 4: Pirfenidone	The animals treated with collagen-polyvinylpyrrolidone and pirfenidone developed less inflammation and fibrosis than animals in the other study groups
Turkmen E et al., 2019 [88]	Pirfenidone	Suppression of TGF-b1 Downregulation of cytokine production Inhibition of fibroblast proliferation	Animal model 14 rats undergoing tracheotomy Group 1: pirfenidone intraperitoneally Group 2: saline solution intraperitoneally	Pirfenidone reduced fibrosis and narrowing of tracheal lumen diameter significantly versus control group
Fan Y et al., 2021 [89]	Nintedanib	Blockage of the autophosphorylation with consequent inhibition of downstream signalling cascades of FGFRs, PDGFRs, VEGFRs	Animal model and In vitro study of human cells Post-surgical model of tracheal stenosis in rats Group 1: nintedanib Group 2: saline Tracheal specimens were harvested after 3 weeks	Nintedanib prevented tracheal stenosis, and reduced collagen deposition, the expression of fibrotic marker proteins and CD4+ T-lymphocyte infiltration
Namba DR et al., 2015 [92]	Rapamycin	Inhibition of the mammalian target of rapamycin (mTOR)	Controlled in vitro study Fibroblasts isolated from biopsies of 5 patients with LTS were cultured and treated with rapamycin or dimethylsulfoxide or normal controls	Rapamycin significantly decreased proliferation, metabolism and collagen deposition of human LTS fibroblasts compared to dimethylsulfoxide or normal controls
Duvvuri M et al., 2019 [93]	Drug-eluting stent containing rapamycin	Inhibition of the mammalian target of rapamycin (mTOR)	Animal model and in vitro study Mice	In vitro, rapamycin stent decreased collagen-1 deposition and fibroblasts cell proliferation In vivo, rapamycin stent reduced lamina propria thickness and collagen 1, collagen-3, TGF-b and a-SMA expression
Mizokami D et al., 2015 [94]	Tacrolimus	Inhibition of calcineurin with suppression of T-cell activation	Animal model 19 rats with acute tracheal injury Group 1: control Group 2: high-dose tacrolimus Group 3: low-dose tacrolimus	Low dose of tacrolimus prevented laryngotracheal stenosis compared to the untreated animals
Dillard DG et al., 2001 [46]	Anti-human neutralizing antibodies to TGF-b1	TGF-b inhibition	Animal model Laryngotracheal injury in rats Group 1: osmotic pump infusion of TGF-b1 Group 2: pump infusion of neutralizing antibodies Group 3: control	TGF-b1 infusion increased the expression of ECM proteins compared to control Neutralizing antibodies decrease ECM protein expression in the in airway
Simpson CB et al., 2008 [47]	Anti-human neutralizing antibodies to TGF-b1	TGF-b inhibition	Animal model Modified canine model of LTS Group 1: saline injection into the injury site Group2: combination of local injection and intravenous anti-TGF-b	Combination of intralesional and intravenous anti-TGF-b resulted in a reduction in tracheal stenosis and an increase survival time compared to control animals

## Data Availability

Not applicable.

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
