# Peer review of "Molecular Mechanisms and Physiological Changes behind Benign Tracheal and Subglottic Stenosis in Adults"

_ijms, 2022, doi:10.3390/ijms23052421_

Round 1
Reviewer 1 Report
Thank you for this comprehensive Review. Please see my comments below
L61 "Laryngotracheal surgery can be considered the first line treatment for LTS, despite the intrinsic limitations related to the surgical procedure itself and/or the patient’s clinical status"
> I would recommend to weaken this sentence as I do not think that surgery is still considered to be a first line treatment everywhere (as you explained afterwards, too)
L66 "The understanding of the pathophysiology of LTS is still an ongoing process, is crucial to develop new more effective therapeutic strategies and to prevent restenosis."
> Think about adding "prevention of LTR" to your arguments.
Fig.1
- missing explanation of some abbreviations caused by mixing up the captions of Fig. 1 and 2.
- unusual arrangement, reducing comprehensibility
- defective typesetting
L525 empty brackets
Fig.2
- See above, captions mixed up
L720 "A new medical therapy.."
> which new medical therapy?
L740 TLS = LTS?
L787 See comment L61
Author Response
Reviewer 1
Thank you for this comprehensive Review. Please see my comments below
We want to thank the Reviewer for the evaluation of our work. We are grateful for the comments and the suggestions made. We have answered to all the comments and remarks in order to improve the manuscript accordingly.
Reviewer 1’s comment 1
L61 "Laryngotracheal surgery can be considered the first line treatment for LTS, despite the intrinsic limitations related to the surgical procedure itself and/or the patient’s clinical status".
> I would recommend to weaken this sentence as I do not think that surgery is still considered to be a first line treatment everywhere (as you explained afterwards, too)
Answer to Reviewer 1’s comment 1
We thank the Reviewer for this important comment. We have modified the manuscript as follows: “Laryngotracheal surgery may be considered an option for LTS, despite the intrinsic limitations related to the surgical procedure itself and/or the patient’s clinical status2”:
Reviewer 1’s comment 2
L66 "The understanding of the pathophysiology of LTS is still an ongoing process, is crucial to develop new more effective therapeutic strategies and to prevent restenosis."
> Think about adding "prevention of LTR" to your arguments.
Answer to Reviewer 1’s comment 2
We thank the Reviewer for this comment. We have modified the whole paragraph by adding the following sentence according to Reviewer suggestion: “The understanding of the pathophysiology of LTS is still an ongoing process and is crucial to develop new more effective therapeutic strategies. Further, the prevention of LTS and its recurrence represents an important clinical need”.
Reviewer 1’s comment 3
Fig.1
- missing explanation of some abbreviations caused by mixing up the captions of Fig. 1 and 2.
- unusual arrangement, reducing comprehensibility
- defective typesetting
Answer to Reviewer 1’s comment 3
We really thank the Reviewer for these comments. We do apologize for the mixing up of figure captions. We have now properly arranged and modified captions in order to increase comprehensibility.
Reviewer 1’s comment 4
L525 empty brackets
Answer to Reviewer 1’s comment 4
We thank the Reviewer for this comment. We have emended the manuscript.
Reviewer 1’s comment 5
Fig.2
- See above, captions mixed up
Answer to Reviewer 1’s comment 5
We thank the Reviewer for this comment. We do apologize for the mixing up of figure captions. We have now properly arranged and modified captions in order to increase comprehensibility.
Reviewer 1’s comment 6
L720 "A new medical therapy.."
> which new medical therapy?
Answer to Reviewer 1’s comment 6
We thank the Reviewer for this comment. We have rephrased the whole paragraph as follows: “. Pharmacological approaches based on antifibrotic medications are currently being tested at pre-clinical level. These treatments may have the potential to open up new prospects for future treatments, aimed at regulating immunological derangement and/or slowing down the deposition of pathologic ECM in LTS”.
Reviewer 1’s comment 7
L740 TLS = LTS?
Answer to Reviewer 1’s comment 7
We thank the Reviewer for this comment. We have emended the manuscript.
Reviewer 1’s comment 8
L787 See comment L61
Answer to Reviewer 1’s comment 8
We thank the Reviewer for this comment. We have modified the sentence as follows: “At present the therapeutic strategies to treat LTS include laryngotracheal resection and reconstruction, even though endoscopic treatment is spreading as a less invasive treatment option”.

Reviewer 2 Report
The manuscript presents a comprehensive review of the molecular and mechanical mechanisms behind the development and pathogenesis of LTS and discusses the potential clinical implications. It has been organized well and written clearly.
Minor comments:
- Figures 1 and 2 have been placed in reverse order. Please also resize Figure 1 to fit the page width. Figures titles and captions should be below the figures and not above them. It is also better to name the subfigures “a” and “b” and refer to them instead of using “Left” and “Right” in the caption.
- Line 525: Please write σ and ε in the parentheses.
- Line 571: The symbol for the dynamic viscosity of a material is “µ” instead of “n” written in the manuscript.
- Line 623: It should be ΔP. Please correct.
Author Response
Reviewer 2
The manuscript presents a comprehensive review of the molecular and mechanical mechanisms behind the development and pathogenesis of LTS and discusses the potential clinical implications. It has been organized well and written clearly.
We want to thank the Reviewer for the evaluation and the appreciation of our work. We are grateful for the suggestions made and the comments made. We have answered to all the comments and remarks in order to improve the manuscript accordingly.
Minor comments
Reviewer 2’s comment 1
Figures 1 and 2 have been placed in reverse order. Please also resize Figure 1 to fit the page width. Figures titles and captions should be below the figures and not above them. It is also better to name the subfigures “a” and “b” and refer to them instead of using “Left” and “Right” in the caption.
Answer to Reviewer 2’s comment 1
We really thank the Reviewer for this comment. We do apologize for the mixing up of figure captions. We have now properly arranged and modified captions in order to increase comprehensibility.
Reviewer 2’s comment 2
Line 525: Please write σ and ε in the parentheses.
Answer to Reviewer 2’s comment 2
We thank the Reviewer for this question that gave us the chance to better specified this point (methods, lines 119-120). The physicians that were in charge of patients were blind to physiological measurement.
Reviewer 2’s comment 3
Line 571: The symbol for the dynamic viscosity of a material is “µ” instead of “n” written in the manuscript.
Answer to Reviewer 2’s comment 3
We thank the Reviewer for this comment. We have emended the manuscript accordingly.
Reviewer 2’s comment 4
Line 623: It should be ΔP. Please correct.
Answer to Reviewer 2’s comment 4
We thank the Reviewer for this comment. We have emended the manuscript accordingly.
